



# Instability of Northeast Siberian ice sheet during glacials

Zhongshi Zhang[1,2,3,4], Qing Yan[4], Elizabeth J. Farmer[5], Camille Li[5], Gilles Ramstein[6], Terence Hughes[7], Martin Jakobsson[8,9], Matt O'Regan[8,9], Ran Zhang[10], Ning Tan[6], Camille Contoux[6], Christophe Dumas[6], Chuncheng Guo[2]

[1]Department of Atmospheric Science, School of Environmental studies, China University of Geoscience, Wuhan, 430074, China
[2]Uni Climate, Uni Research AS and Bjerknes Centre for Climate Research, 5007 Bergen, Norway
[3]Center for Early Sapiens Behaviour, 5007 Bergen, Norway
[4]Nansen-Zhu International Research Center, Institute of Atmospheric Physics, Chinese Academy of Sciences, 100029, Beijing, China
[5]Geophysical Institute, University of Bergen and Bjerknes Centre for Climate Research, 5007 Bergen, Norway
[6]Laboratoire des Sciences du Climat et de l'Environnement, LSCE/IPSL, CEA-CNRS-UVSQ, Université Paris-Saclay, F-91191 Gif-sur-Yvette, France
[7]School of Earth and Climate Sciences, Climate Change Institute, University of Maine, Orono, Maine 04469, USA
[8]Department of Geological Sciences, Stockholm University, 10691, Stockholm, Sweden
[9]Bolin Centre for Climate Research, Stockholm University, 10691, Stockholm, Sweden
[10]Climate Change Research Center, Chinese Academy of Sciences, Beijing 100029, China

*Correspondence to*: Zhongshi Zhang (zhongshi.zhang@uni.no) and Qing Yan (yanqing@mail.iap.ac.cn)

**Abstract.** It has been widely believed that Northeast (NE) Siberia remained ice-free during most Pleistocene Northern Hemisphere (NH) glaciations, while ice sheets extended gradually across North America and Northwest (NW) Eurasia. However, recent fieldwork has provided robust evidence of ice sheets occupying the shallow continental shelf of the East Siberian Sea during several Pleistocene glaciations. The debate surrounding the existence and history of this enigmatic NE Siberian ice sheet highlights fundamental gaps in our current understanding of the mechanisms of glacial climate evolution. Here, we combine climate and ice sheet simulations to demonstrate how ice-vegetation-atmosphere-ocean dynamics can lead to two ice sheet configurations: the well-known Laurentide-Eurasian configuration with large ice sheets over North America and NW Eurasia, and a circum-Arctic configuration with large ice sheets over NE Siberia and the Canadian Rockies. Compared to the Laurentide-Eurasian configuration, formation of the circum-Arctic configuration can occur with an atmospheric stationary wave pattern similar to today's. Once the circum-Arctic configuration is established, it amplifies atmospheric stationary waves, leading to surface warming in the North Pacific, ablation of the NE Siberian ice sheet, and ultimately a swing to the Laurentide-Eurasian configuration. Our simulations highlight the complexity of glacial climates, and may hint towards potential mechanisms for interglacial-glacial transitions.



## 1 Introduction

Today, the only remaining large-scale ice sheet in the NH exists on Greenland. During the Last Glacial Maximum (LGM, ~21 kyr ago) ice sheets expanded down to ~50 $^{o}$N across North America and NW Eurasia (Fig.1). In North America, the Hudson Bay region was covered by the Laurentide ice sheet, with further ice sheets across the Canadian Arctic Archipelago (Innuitian ice sheet) (Dyke et al., 2002) and the Canadian Rockies (Cordilleran ice sheet) (Dyke, 2004). In NW Eurasia, ice sheets covered the Barents Sea and part of the Kara Sea (Barents ice sheet) (Svendsen et al., 2004), Fennoscandia (Fennoscandian ice sheet) (Hughes et al., 2016), Iceland (Geirsdottir, 2011) and large parts of the British Isles and Ireland (Svendsen et al., 2004). These ice sheets, in particular the Laurentide ice sheet, strongly influenced atmospheric stationary waves (e.g., Cook and Held, 1988), creating a stronger, more zonal jet stream (e.g., Li and Battisti, 2008; Hofer et al., 2012; Ullman et al., 2014), weaker storm tracks (e.g., Li and Battisti, 2008; Rivière et al., 2010), and altered patterns of wave-breaking and internal atmospheric variability (e.g., Li and Battisti, 2008; Ullman et al., 2014). In contrast, NE Siberia, although located north of 60 $^{o}$N, was portrayed as largely ice-free in most ice-sheet reconstructions (Pitulko et al., 2004; Gualtieri et al., 2005; Abe-Ouchi et al., 2013; Kleman et al., 2013; Peltier et al., 2015).

The history of the NE Siberian ice sheet has long been discussed. Although geological (Gualtieri et al., 2005) and archaeological (Pitulko et al., 2004) evidence demonstrates that NE Siberia was ice-free just before or during the LGM, glacial landforms on the continental margin and other studies in the region show a NE Siberian ice sheet was, at some point(s) in time, extensive (Niessen et al., 2013; Jakobsson et al., 2016; O'Regan et al., 2017; Nikolskiy et al., 2017). Reconstructions show it is possible to grow ice sheets over NE Siberia during the last (Charbit et al., 2007; Liakka et al., 2016) and penultimate glacials (Colleoni et al., 2016; Wekerle et al., 2016). However, due to no evidence supporting NE Siberian ice sheet during the LGM, most syntheses of NH ice sheets of the last a few glacials do not include a NE Siberian ice sheet (Abe-Ouchi et al., 2013; Kleman et al., 2013; Peltier et al., 2015) or attribute its presence to model bias (Charbit et al., 2007). It is believed that the presence of the Laurentide and Fennoscandian ice sheets (Liakka et al., 2016), or increased snow-free days due to decreased dust-mixed snow albedo (Krinner et al., 2006) can limit the potential for ice accumulation over NE Siberia.

Here, we investigate the mechanism behind the waxing and waning of the NE Siberian ice sheet, using an asynchronous coupling method, to simulate NH ice sheet growth under glacial conditions forced with constant orbital parameters and greenhouse gas levels. This method has unprecedented advantages in highlighting internal climate feedbacks.

The paper is structured as follows: Section 2 introduces the models and the asynchronous coupling method used in this study; Section 3 presents the simulated results; Section 4 discusses the uncertainties in the asynchronous coupled simulations, geological evidence for NE Siberian ice sheet, and the implications for Quaternary NH ice sheet evolution.

## 2 Model and experimental design

In this study, we simulate a glacial climate firstly using the low-resolution Norwegian Earth System Model (NorESM-L). We use the resulting climatology (temperature, precipitation) to force the BIOME4 equilibrium vegetation model to generate vegetation, and the Parallel Ice Sheet Model (PISM) to generate NH ice sheets. Finally, we use the newly generated NH vegetation and ice sheets as boundary conditions in NorESM-L to simulate a new climate state. This process is repeated six times to see how the ice sheets and climate evolve. To assess uncertainties, we carried out sensitivity atmosphere-only experiments using the high resolution Community Atmosphere Model, CAM4, and repeated the ice sheet simulations using the GRenoble Ice-Shelf and Land-Ice model, GRISLI.

### 2.1 Introduction to the models

Here, we introduce the NorESM, BIOME4 and PISM models. We also present the experimental design details for the NorESM-BIOME4-PISM asynchronous coupled simulations, the sensitivity experiments with an atmosphere-only model and PISM. For an introduction to GRISLI, the other sensitivity experiments and the repeated GRISLI ice-sheet simulations, please refer to the supplement.

The well documented NorESM-L is a state-of-the-art high complexity earth system model (Zhang et al., 2012; Bentsen et al., 2013). It was developed for paleoclimate simulations at the Bjerknes Centre for Climate Research (BCCR), Norway. NorESM-L couples the Miami Isopycnic Coordinate Ocean Model (MICOM) and the spectral Community Atmosphere Model (CAM4) (e.g., Eaton, 2010). The resolution of the ocean is approximately 3° (g37) in the horizontal and 32 layers in the vertical. The resolution of spectral CAM4 is approximately 3.75° (T31) in the horizontal and 26 levels in the

vertical (Eaton, 2010). NorESM-L performs well in simulating the pre-industrial climate (Zhang et al., 2012), and has good

skill in simulating paleoclimates (e.g., Zhang et al., 2012, 2014). Furthermore, the atmospheric component, CAM4, can be

run at different resolutions and in different configurations. The default high-resolution for CAM4 is 0.9° latitude by 1.25°

longitude (F09), and 26 levels in the vertical (Eaton, 2010). At this resolution, CAM4 uses a Finite-Volume (FV) dynamical

5    core (Eaton, 2010), with a conservative flux-form semi-Lagrangian scheme for horizontal discretization and a Lagrangian

vertical coordinate (Eaton, 2010). CAM4 successfully simulates the NH trough and ridge system, in agreement with

observations.

BIOME4 is an equilibrium biogeography model (Kaplan et al., 2003), widely used in simulations of equilibrated

vegetation in past and future climate projections. The model uses the different bioclimatic limits (temperature resistance,

10    moisture requirement and sunshine amount) among plant functional types to simulate the potential natural vegetation of a

given climate. It uses a horizontal resolution of 0.5° latitude by 0.5° longitude and simulates the equilibrium distribution of

28 biomes. The input conditions for the model include atmospheric $CO_2$ concentration, soil physical properties (water-

holding capacity and percolation rate) and climate forcings, including temperature, precipitation, sunshine and minimum

temperature.

15    PISM is a three-dimensional, thermodynamically coupled continental-scale ice sheet model (Martin et al., 2011;

Winkelmann et al., 2011), widely used in ice sheet modelling (e.g., Yan et al., 2016; Bakker et al., 2017). It is based on the

shallow ice approximation (SIA) and the shallow shelf approximation (SSA). Ice velocity is the sum of the velocities from

the SIA and the SSA, providing a consistent treatment for different flow regimes in ice sheets, streams, and shelves. Surface

mass balance is calculated as the difference between snowfall accumulation and surface melting. The snowfall is the

20    partitioned precipitation according to the empirical relationship between precipitation and temperature. Surface melting is

estimated according to the positive degree-day scheme. Here, the default melt rate is 8 mm/d$^o$C for ice (PDD_ice), and 3

mm/d$^o$C for snow (PDD_snow), with a standard deviation 5 $^o$C of surface air temperature (Temp_std). The melted snow is

able to refreeze as superimposed ice. PISM includes an ocean model to calculate basal melt rate and temperature at the base

of the ice shelf. It calculates the position of the grounding line by using the flotation criterion. The model also employs a

25    physically based 2D-calving parameterization and a thickness limitation calving mechanism. It uses the pseudo-plastic





power law to estimate basal sliding. More detailed PISM introductions can be found in Winkelmann et al. (2011), Martin et al. (2011) and the model manual. Here PISM runs at a resolution of 40 km×40 km.

## 2.2 Experimental design for NorESM-BIOME4-PISM asynchronous coupled simulations with default PISM parameters

5       We use an asynchronous coupling method to carry out our simulations (Table 1). In the first step, we push NorESM-L to simulate an idealized glacial climate by changing the Earth's obliquity to 22 degrees (minimum of last four glacial-interglacial cycles) and atmospheric $CO_2$ level to 200 ppmv. Earth's eccentricity and precession are fixed at values of year 1950. We then use this simulated cold climate to force BIOME4 and PISM to generate the first vegetation and ice sheet simulations. We take the simulated tundra, taiga, and ice sheet configurations and put them back into NorESM-L to get a

new cold climate, forced with the same glacial obliquity and atmospheric $CO_2$ level. We return this back into BIOME4 and PISM to get the new vegetation and ice sheet simulations. This iterative process, with the evolving NorESM-L glacial climate and BIOME4 and PISM vegetation and ice sheet configurations, was repeated six times in total. Our idealized glacial climate, forced by constant obliquity and atmospheric $CO_2$ levels, highlights ice-atmosphere-ocean feedbacks. These previous studies are based on reconstructed climate conditions (Abe-Ouchi et al., 2013), and include - at best - only crude

versions of the important feedbacks. Compared to Earth-system Models of Intermediate Complexity (Ganopolski et al., 2010; Beghin et al., 2014), high complexity climate models provide advantages in simulating interactions between ice sheets and atmospheric stationary waves (e.g., Cook and Held, 1988), Atlantic jet streams (e.g., Hofer et al., 2012; Ullman et al., 2014), storm tracks and variability patterns (e.g., Li and Battisti, 2008; Rivière et al., 2010).

       We include two group simulations in this study (Table 1). In the first group, we use the modern land-sea

distributions. In the second group, we change the Barents Sea into land in order to get better simulations for the Barents ice sheet. We run the control (Pi and Pb) and spin up (Pi_GlcI and Pb_GlcI) glacial experiments for 2200 years and use the last 200 years. We run all other coupled NorESM-L experiments for 500 years and use the last 100 years.

       With BIOME4 we use an anomaly method to simulate vegetation under glacial conditions. We firstly calculate anomalies between the cold climate and the modern control run, including monthly mean temperature, monthly precipitation,

monthly mean sunshine and annual minimum temperature. Then we add these anomalies onto the default BIOME4 modern



climatology, to create the climate forcing for each BIOME4 experiment. For example, we add the anomalies between PiGlc_V and Pi on the default BIOME4 modern climatology to create the climate forcing for the B_PiGlcV experiment (Table 1). In all BIOME4 experiments, atmospheric $CO_2$ concentration is set to 200 ppmv, and soil water-holding capacity and percolation rate remain identical to present conditions.

With PISM, we input the NorESM-L simulated atmosphere surface temperature (SAT), precipitation, ocean temperature and salinity to get the ice sheet simulations. We interpolate the simulated NorESM-L temperature and precipitation to the resolution of PISM (40 km×40 km) for the NH. Then we perform a topography correction on the interpolated temperature and precipitation, to account for the topography difference between NorESM-L and PISM. The modern topography for PISM comes from the ETOPO1 datasets. For the temperature correction, we employ a uniform lapse-
rate correction γ, which equals 7 °C/km. For the correction on precipitation, we use an empirical law, in which every 1°C of temperature change leads to a 5.1% precipitation change. We use an anomaly method to create ocean temperature and salinity for the PISM ocean component. We calculate the anomalies of temperature and salinity averaged between 0 and 200 m in a simulated glacial climate, relative to its control experiment. Then we add the anomalies onto the PISM modern ocean temperature and salinity climatology, which is derived from the World Ocean Atlas 2013 version 2 (1975~2004) for the NH.
In each step, PISM is initialized from the ice sheet configuration from the preceding step. For example, P_PiGlcV is initialized from the P_PiGlcIV ice sheet configuration. All PISM experiments are integrated for 100 kyr to reach quasi-equilibrium.

### 2.3 Sensitivity experiments with CAM4

      In order to investigate the feedbacks behind the simulated ice sheet configurations and climate, we choose two
kinds of NH ice sheet (Fig. S1) and three SST fields (Fig. S1) to carry out atmosphere-only sensitivity experiments with the high-resolution CAM4. In total, we design six atmosphere only simulations (Table 2).

### 2.4 Sensitivity experiments with PISM

      In order to test the influence of parameter uncertainties in the simulated ice sheets, we choose three other sets of parameters (Min, Max, Maxi, Table 3), in addition to the default set. We test the three parameters for surface mass balance –

PDD_ice, PDD_snow and SAT_std – and two important parameters for ice dynamics – ENF_SIA (the SIA enhancement factor) and ENF_SSA (the SSA enhancement factor). These selected parameters are prevalent in other modelling studies to test the sensitivity of ice sheets (e.g., Stone et al., 2010; Yan et al., 2016). Larger values of these five parameters lead to smaller ice sheet extents (due to enhanced surface melting and ice discharge), and vice versa.

For each simulated glacial climate, we further carry out sensitivity PISM experiments with these three sets of parameters. For example, with the same climate forcing simulated in PiGlc_V, we carry out PISM experiments P_PiGlcV_Min, P_PiGlcV_Max, and P_PiGlcV_Maxi, in addition to the default P_PiGlcV. In this way, in total we carry out 24 PISM sensitivity experiments. All these experiments are integrated to reach quasi-equilibrium.

## 3 Results

Two distinct NH ice sheet configurations (Fig. 2, Fig. S2 and S3) appear in our simulations. The first is the circum-Arctic ice sheet configuration (Fig. 2a), with a large NE Siberian ice sheet connecting with the Cordilleran ice sheet and smaller and lower Laurentide and NW Eurasian ice sheets. The second is the Laurentide-Eurasian ice sheet configuration (Fig.2c), with large Laurentide and NW Eurasian ice sheets, a limited Cordilleran ice sheet, and no NE Siberian ice sheet. In this configuration an ice-free passage links Siberia and North America via the Bering land bridge.

The important feature of our simulations is the swing from the circum-Arctic to the Laurentide-Eurasian ice sheet configuration. When forced with the circum-Arctic ice sheet configuration, NorESM-L simulates an altered atmospheric circulation with amplified stationary waves (Fig. 2b), which results in PISM producing the Laurentide-Eurasian ice sheet configuration. Alternatively, when forced with the Laurentide-Eurasian configuration, the simulated circulation with weaker stationary waves (Fig. 2d) results in PISM producing the circum-Arctic ice sheet configuration. This suggests that no change

in external forcing is required - changes in atmospheric circulation resulting from changing ice sheets alone can push one configuration into the other.

Our coupled and atmosphere-only experiments demonstrate that ice-atmosphere-ocean feedbacks (Cook and Held, 1988) generate these swings in ice sheet configurations. When the circum-Arctic configuration is established, the presence of the NE Siberian and Cordilleran ice sheets alters the atmospheric circulation, leading to an enhanced stationary ridge over

NE Siberia and the Canadian Rockies (Fig. 2b and Fig. S4). Over North America and NW Eurasia, the deepened troughs

promote colder temperatures (Fig. S5) and increased snow cover (Fig. S6). Meanwhile, the enhanced ridge creates

anomalous anticyclonic winds over the North Pacific, bringing more warm surface waters and heat northward. The resulting

regional warming exacerbates the melting of the NE Siberian and Cordilleran ice sheets (Fig. 3), and there is a swing

towards the Laurentide-Eurasian configuration. In contrast, the Laurentide-Eurasian configuration produces an atmospheric

circulation with relatively weak stationary waves (Fig. 2d and Fig. S4).  The remote effects of changing ice sheet elevation

(such as responses to surface winds, temperature and ocean circulation) have been highlighted in previous studies (Ullman et

al., 2014; Zhang et al., 2014), and suggest that a growing Laurentide ice sheet could limit the expansion of its Eurasian

counterpart (Liakka et al., 2016), and thus help promote a swing to the circum-Arctic configuration.

Our ice sheet sensitivity experiments show that the evolution of the circum-Arctic configuration has a timescale of

tens of thousands years. Although choosing parameters for ice sheet models unavoidably includes uncertainties, the swings

and the timescales involved are independent of the selected PISM parameters (Fig. S7 and S8), and indeed, the ice sheet

model used (results are repeatable with GRISLI, Fig. S9). In sensitivity experiments sampling a range of parameters, PISM

reaches quasi-equilibrium very quickly under constant climate forcings (Fig. 4 and Fig. S7), suggesting a ~100 kyr glacial is

long enough to include a number of swings in ice sheet configurations.

## 4 Discussion and summary

### 4.1 Geological evidence for large NE Siberian ice sheet

The orientation of glaciogenic features mapped on the NE Siberian continental shelf (Niessen et al., 2013;

Jakobsson et al., 2016), the recently discovered glacially scoured trough on the outer margin north of the De Long Islands

(O'Regan et al., 2017), and glacial deposits on the New Siberian Islands (Nikolskiy et al., 2017) provide robust evidence for

a NE Siberian ice sheet. Due to the difficulty of dating glacial deposits, it is not easy to firmly constrain the exact timing(s)

of these glacial activities, though evidence (Jakobsson et al., 2016; Nikolskiy et al., 2017; Colleoni et al., 2016; Wekerle et

al., 2016) suggests that Marian Isotope Stage 6 (MIS6) (180-140 ka) in particular saw an extensive ice sheet in NE Siberia.

This does not preclude the development of a less extensive ice sheet in NE Siberia after this time. While the glacial diamicts



overlying grounding zone features in the De Long Trough are older than the LGM, they could, given the existing radiocarbon age constraints, be as young as MIS4 (~74-60 kyr) (O'Regan et al., 2017). In NE Siberia, a similar sedimentological environment not only appeared in MIS4 and MIS6, but also some in earlier MISs (Melles et al., 2007).

The mineral and isotopic composition of ice rafted debris (IRD) is another clue for determining the dynamics of circum-Arctic ice sheets. Arctic IRD records are traditionally interpreted in terms of competing inputs between Eurasian and North American ice sheets (Spielhagen et al., 2004). More recent studies have identified mineral and isotopic signatures of IRD from an NE Siberian provenance (Fagel et al., 2014; Kaparulina et al., 2016; Dong et al., 2017). Notably, on the Lomonosov Ridge, dolomite rich IRD (originating from the North American continent) dominated the MIS6-5 transition, while IRD from the Laptev and East Siberian Sea regions dominated MIS4 (Kaparulina et al., 2016). Similar shifts occurred in the Canada Basin, north of the Chukchi Plateau (Dong et al., 2017), where the mineral composition of IRD shows a bias towards NE Siberian sources during MIS4, but towards sources from the Laurentide Ice Sheet later in MIS3. Although considerable work is needed to investigate frequent shifts in IRD sources during glacials, and the mechanism behind, the broad pattern of IRD inputs during the last glacial cycle conforms to our simulated swings in ice sheet configurations. Our simulations suggest that, in addition to the varied surface ice drift system (Bischof et al., 1997), relatively rapid, asynchronous growth and decay of the circum-Arctic ice sheets may regulate these shifts in IRD during glacial cycles.

Further afield, IRD records from sub-Arctic regions also display contrasting source patterns between NE Siberia and North America. When IRD content is high in the marginal seas of NE Siberia, e.g. Okhotsk Sea (Nürnberg et al., 2011), IRD content in the North Atlantic (i.e. from a Laurentide source) is low (Hodell et al., 2008); and vice versa. Furthermore, during MIS 4 and MIS 6, sea level drops were in the order of 50-70 and 100 m respectively (Rohling et al., 2014). As simulated in our experiments, which show ice volumes equal to sea level drop in the order (Fig. S2 and S3), a substantial NE Siberian ice sheet would be required.

Our simulations present new insights with which to reconsider the complex and controversial array of both glaciated and unglaciated evidence in NE Siberia. Assessing glacial geological evidence from NE Siberia is complicated by permafrost, regolith and basal topography processes that would have occurred beneath the ice sheet itself (Hughes, 1973). Geological mapping only presents the evidence from thawed areas of surface deposition and erosion, and this is only a tiny

fraction of what exists at depth. Postglacial thermokarst processes have also transformed regolith permafrost so drastically that glacial features would be difficult to recognize even at depth, when that geology becomes accessible. However, the wide-spread moraine sediments in NE Siberia (Barr and Clark, 2012) show clear age contrasts between their western and eastern limits, highlighting an area of study with huge potential for assessing the full extent of the NE Siberian ice sheet and

its pattern of retreat.

## 4.2 Implications for Quaternary NH ice sheet evolution

The complex feedbacks presented here highlight the complexities of the NH glacial climate system. Changes in atmospheric stationary waves likely played an important role in large climate transitions, particularly when changes in external forcings (e.g. insolation) are not large enough to explain the climate response – e.g. during glacial inception. When

only forced with orbital parameters and greenhouse gas levels, changes in atmospheric circulation are weak (Hofer et al., 2012) (Fig. S10). However, in a glacial climate, vegetation feedbacks seem to be important for the formation of the circum-Arctic configuration. For example, the large growth of ice sheets in P_PiGlcII (Fig. S2f) and P_PbGlcII (Fig. S3f) relative to P_PiGlcI (Fig. S2d) and P_PbGlcI (Fig. S3d) indicates the expansion of tundra during glacials is an important condition for ice sheet formation in NE Siberia. Once ice has formed, it is the presence of the ice sheets themselves that becomes the

dominant factor (Cook and Held, 1988; Hofer et al., 2012) and, as our study highlights, triggers ice-atmosphere-ocean feedbacks and generates the swings in NH ice sheet configurations in a full glacial. Our study suggests that during some glacials ice sheet expansion starts from a circum-Arctic configuration, rather than a gradual expansion into the Laurentide-Eurasian configuration, as is often assumed.

In reality, climate forcings evolve alongside the ice sheet so, in an ideal world, fully coupled transient simulations

(Ganopolski et al., 2010; Beghin et al., 2014) would most reliably mimic the interaction between climate and ice sheet. However in a transient simulation, due to the complexity of a fully coupled system, it is quite difficult to distinguish the importance of orbital forcings, greenhouse gas levels and internal ice-vegetation-atmosphere-ocean feedbacks. With constant climate forcing, our asynchronous simulations provide a wealth of new and valuable information, which represents the first step in beginning to fully assess the question of the internal feedbacks and ice sheet configurations. Whilst climate and ice

sheet interactions may influence the resulting ice sheet volume and shape, they are unlikely to change whether or not an ice

sheet actually forms in NE Siberia or North America. Once developed, ice sheets tend to continue to grow during a full

glacial, so the NE Siberian or Laurentide ice sheets should do exactly that, until they become large enough to trigger a swing

to another ice sheet configuration. Seen in this way, before a fully coupled ocean-atmosphere-vegetation-ice sheet high

complexity model (with a good ability to simulate atmospheric stationary waves) is really available for transient glacial-

interglacial simulations, asynchronous coupled simulations remain a good choice to reveal the importance of atmosphere-

ocean-vegetation-ice sheet feedbacks in the swings in ice sheet configurations.

When we consider both our new modelling results and the existing geological evidence for the history of the NE

Siberian ice sheet, we believe the established idea of a gradual expansion into the Laurentide-Eurasian ice sheet

configuration has to be revaluated. Our simulations neglect some interactions between the ice sheet and climate variabilities

on an orbital timescale, since we run the model to equilibrium with a fixed climate forcing at each step. It still raises some

fundamental questions about glacial climate, not least: why was NE Siberia glaciated during some glacials and not others?

The internal atmosphere-ocean-vegetation-ice sheet feedbacks highlighted in this study seem to be the key to answer this

question. How orbital forcings influence the internal feedbacks to lead to the presence or absence of the NE Siberian ice

sheet in past glacials is a question for the future as the possibilities for running transient simulations.

**4.3 Summary**

Our study highlights the importance of ice-vegetation-atmosphere-ocean feedbacks in NE Siberian ice sheet

evolution during glacials. Previous transient modelling studies (Abe-Ouchi et al., 2013; Ganopolski et al., 2010; Beghin et

al., 2014) were carried out with interpolated climate forcings based on isotopic records, or interpolated climate forcings from

several snapshot GCM simulations, or Earth-system Models of Intermediate Complexity (EMICs). These transient

simulations although include - at best - only crude versions of the important feedbacks, they neglect to consider the existence

and instability of the NE Siberian ice sheet. Coupled models, such as the one used here, have better representations of these

feedbacks, but are too computationally expensive to be used in transient simulations for full glacial-interglacial cycles. The

swings in ice sheet configurations we see seem to have a timescale that would fit within the prolonged ~100 kyr glacials after

the Mid-Pleistocene transition (~1 Ma). In each ~100 kyr glacial cycle, the development of NH ice sheets is likely to be

different due to the ice-vegetation-atmosphere-ocean feedbacks trigged by different combinations in Earth orbital parameters



and greenhouse gas levels. Establishing whether a large NE Siberian ice sheet existed during the early stages of last glacial requires more robust glaciogeological and chronologic evidence. This would indicate that the gradual build-up of in the Laurentide ice sheet later played a role in abrupt events such as the Dansgaard-Oeschger and Heinrich events of the last glacial (Hofer et al., 2012; Colleoni et al., 2016; Beghin et al., 2014). Moreover, the alternating ice sheet configurations

would have left NE Siberia almost ice free, potentially playing a role in human global dispersal (Goebel et al., 2008; Timmermann and Friedrich et al., 2016) by allowing prehistoric humans to cross the Bering land bridge.

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

**Acknowledgments**

This study was jointly supported by the National Natural Science Foundation of China (Grant No. 41472160), the Strategic Priority Research Program of the Chinese Academy of Sciences (Grant No. XDA19070402), the Norwegian Research

Council (Project No. 221712 and 229819), and the NordForsk-funded project GREENICE (Project No. 61841). The paper is in memory of the supercomputer Hexagon in Norway. All atmosphere-only simulations are carried on the new cluster in the Department of Atmospheric Science, CUG.



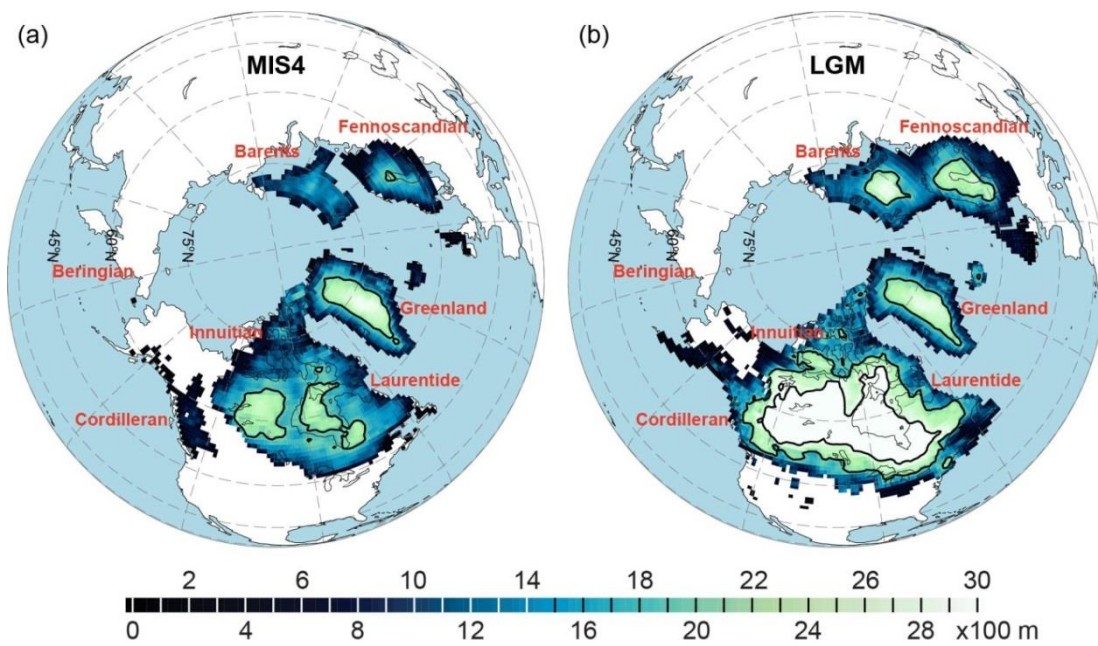

5    **Fig. 1. ICE-6G (VM5a) ice sheet reconstructions (Peltier et al., 2015).** (a) Marine isotope stage 4 (MIS4, ~74 kya) and (b) LGM (~21 kya).







**Fig. 2**. **Alternating cycle in two ice sheet configurations.** (a) A circum-Arctic ice sheet configuration, causes (b) changes in near-surface temperature (shaded, $^{o}$C) and annual mean 500hPa geopotential height (red line), leading to a swing to (c) Laurentide-Eurasian ice sheet configuration. The Laurentide-Eurasian ice sheet configuration, causes (d) changes in near-surface temperature (shaded, $^{o}$C) and annual mean 500hPa geopotential height (red line), favouring a swing to the circum-Arctic ice sheet configuration. The two ice sheet configurations (a) and (c) illustrated here are the two PISM experiments P_PbGlcIV and P_PbGlcV in Group 2. The two climate response panels (b) and (d) are two NorESM-L experiments PbGlc_IV and PbGlc_V. The grey line is the annual mean 500hPa geopotential height from the glacial reference experiment PbGlc_II. See more simulations in Fig. S2 and S3.





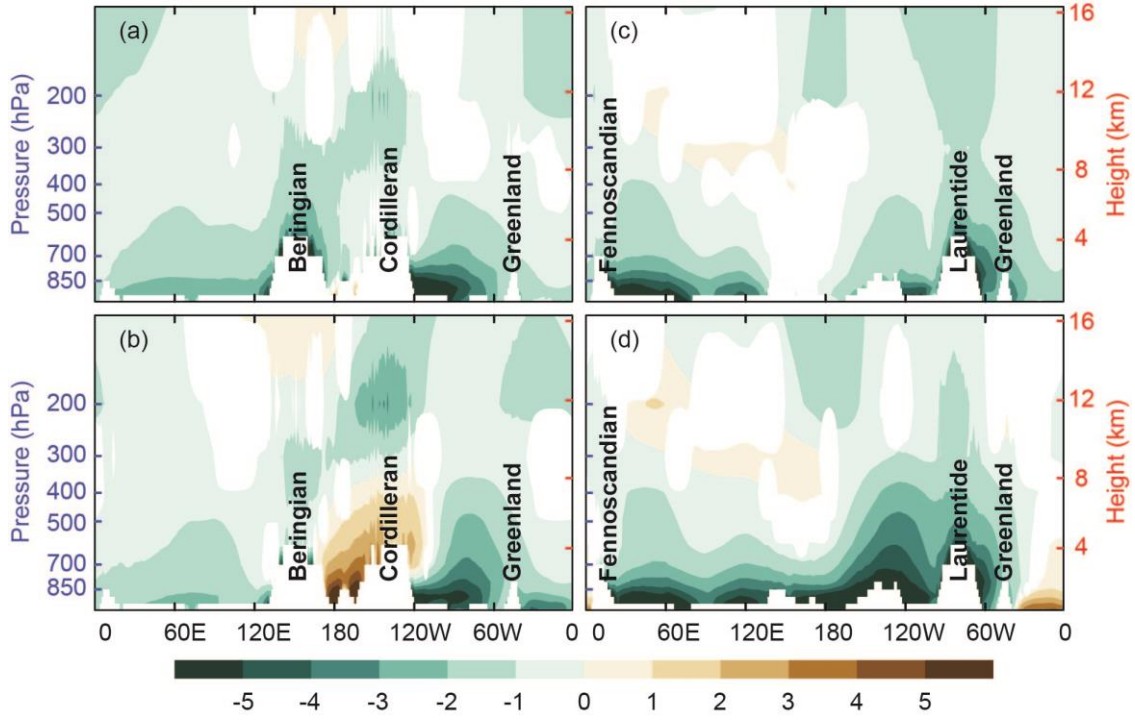

**Fig. 3**. **Atmospheric temperature changes over ice sheets simulated with CAM4.** Changes in annual mean temperature (°C) along ~64°N, caused by a circum-Arctic ice sheet configuration (a) without changes in SST forcing and (b) together with warming of the North Pacific SST, caused by a Laurentide-Eurasian ice sheet configuration (c) without changes in SST forcing and (d) together with cooling of the North Pacific SST. See boundary conditions in Fig. S1.

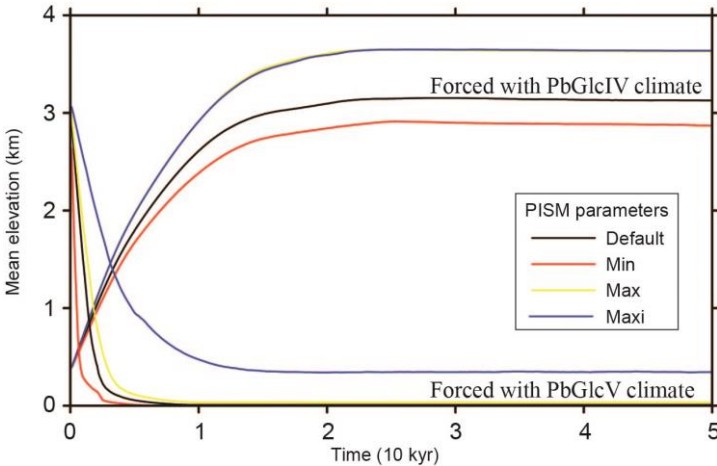

**Fig. 4**. **Time series of ice thickness forced with four sets of PISM parameters.** The ice thickness is averaged over the quadrilateral area marked in Fig. 2. The four lines with an increasing (decreasing) trend are forced with PbGlcIV (PbGlcV) climate, but with different PISM parameters. The black line with the increasing (decreasing) trend indicate how the circum-Arctic (Laurentide-Eurasian) ice sheet configuration shown in Fig. 2 comes into being, starting from the Laurentide-Eurasian (circum-Arctic) ice sheet configuration in the previous step. See more simulations in Fig. S7.



**Table 1. Experimental design for NorESM-BIOME4-PISM experiments with default PISM parameters.**

| Conditions | NorESM | Biome4 | PISM | GRISLI |
|---|---|---|---|---|
| *Group 1: Modern land-sea distribution* | | | | |
| Obliq. 23.4462 °, $CO_2$ 280 ppmv | Pi | | P_Pi | |
| Obliq. 22 °, $CO_2$ 200 ppmv | PiGlc_I | B_PiGlcI | P_PiGlcI | |
| Obliq. 22 °, $CO_2$ 200 ppmv, B_PiGlcI veg., P_PiGlcI ice | **PiGlc_II** | B_PiGlcII | P_PiGlcII | G_PiGlcII |
| Obliq. 22 °, $CO_2$ 200 ppmv, B_PiGlcII veg., P_PiGlcII ice | PiGlc_III | B_PiGlcIII | P_PiGlcIII | G_PiGlcIII |
| Obliq. 22 °, $CO_2$ 200 ppmv, B_PiGlcIII veg., P_PiGlcIII ice | PiGlc_IV | B_PiGlcIV | P_PiGlcIV | G_PiGlcIV |
| Obliq. 22 °, $CO_2$ 200 ppmv, B_PiGlcIV veg., P_PiGlcIV ice | PiGlc_V | B_PiGlcV | P_PiGlcV | G_PiGlcV |
| Obliq. 22 °, $CO_2$ 200 ppmv, B_PiGlcV veg., P_PiGlcV ice | PiGlc_VI | B_PiGlcVI | P_PiGlcVI | G_PiGlcVI |
| *Group 2: Barents Sea changed to land* | | | | |
| Obliq. 23.4462 °, $CO_2$ 280 ppmv | Pb | | P_Pb | |
| Obliq. 22 °, $CO_2$ 200 ppmv | PbGlc_I | B_PbGlcI | P_PbGlcI | |
| Obliq. 22 °, $CO_2$ 200 ppmv, B_PbGlcI veg., P_PbGlcI ice | **PbGlc_II** | B_PbGlcII | P_PbGlcII | G_PbGlcII |
| Obliq. 22 °, $CO_2$ 200 ppmv, B_PbGlcII veg., P_PbGlcII ice | PbGlc_III | B_PbGlcIII | P_PbGlcIII | G_PbGlcIII |
| Obliq. 22 °, $CO_2$ 200 ppmv, B_PbGlcIII veg., P_PbGlcIII ice | PbGlc_IV | B_PbGlcIV | P_PbGlcIV | G_PbGlcIV |
| Obliq. 22 °, $CO_2$ 200 ppmv, B_PbGlcIV veg., P_PbGlcIV ice | PbGlc_V | B_PbGlcV | P_PbGlcV | G_PbGlcV |
| Obliq. 22 °, $CO_2$ 200 ppmv, B_PbGlcV veg., P_PbGlcV ice | PbGlc_VI | B_PbGlcVI | P_PbGlcVI | G_PbGlcVI |

**Table 2. Sensitivity experiments with CAM4**

| CAM4 | Ice-sheet | SST | Others |
|---|---|---|---|
| *NE Siberian (Beringian) and Cordilleran ice sheets* | | | |
| BerCor_con | Modern ice-sheets | From PbGlc_IV (Fig. S1c) | |
| BerCor_ice | BerCor ice anomalies (Fig. S1a) | From PbGlc_IV (Fig. S1c) | Obliq. 22 ° |
| BerCor_ice&sst | BerCor ice anomalies (Fig. S1a) | From PbGlc_V (Fig. S1d) | $CO_2$ 200 ppmv |
| *Laurentide and Fennoscandian ice sheets* | | | |
| LauFen_con | Modern ice-sheets | From PbGlc_V (Fig. S1d) | |
| LauFen_ice | LauFen ice anomalies (Fig. S1b) | From PbGlc_V (Fig. S1d) | Obliq. 22 ° |
| LauFen_ice&sst | LauFen ice anomalies (Fig. S1b) | From PbGlc_VI (Fig. S1e) | $CO_2$ 200 ppmv |

**Table 3. Parameters for sensitivity PISM experiments**

| | PDD_ice | PDD_snow | Temp_std | ENF_SIA | ENF_SSA |
|---|---|---|---|---|---|
| Default | 8 mm/d°C | 3 mm/d°C | 5 °C | 3 | 0.5 |
| Min | 20 mm/d°C | 8 mm/d°C | 5 °C | 5 | 1 |
| Max | 8 mm/d°C | 3 mm/d°C | 2 °C | 1 | 0.1 |
| Maxi | 3 mm/d°C | 1 mm/d°C | 2 °C | 1 | 0.1 |