# Peer review of "Instability of Northeast Siberian ice sheet during glacials"

_Climate of the Past, 2018_

## Short Comment (SC1) · 3 Aug 2018

**Interactive comment:** Zhang et al. (2018): Instability of Northeast Siberian ice sheet during glacials, Clim. Past Discussions

This paper highlights the potential importance of vegetation and stationary-wave feedbacks on the spatio-temporal evolution of ice sheets during glacial cycles. The vegetation feedback is often omitted in climate modeling studies as reliable reconstruction for glacial condition remain elusive --- e.g. preindustrial vegetation is specified for the LGM in the PMIP1-4 boundary conditions --- making this study somewhat unique.

The stationary-wave feedback, on the other hand, has been extensively studied in recent years, but there is an apparent lack of relevant references on this subject throughout the manuscript, and the few references that are included are often old and/or used in the wrong context. In addition, several recent studies have discussed similar topics as examined here, rendering the novelty of this manuscript somewhat reduced.

Lastly, some of the conclusions presented are both unfounded and highly artificial as they are (undoubtedly) a direct result of the idealized experiment design.

We hope that the questions and comments raised here will be addressed if the manuscript is accepted for publication.

Marcus Lofverstrom and Johan Liakka

###################################################################################

**Specific comments:**

Page 2, lines 8-12 and elsewhere:

**Stationary waves:**
Although Cook and Held (1988) investigated interactions between ice sheets and stationary waves, this is arguably not the most appropriate reference on this topic (at least not mentioned alone as is currently the case). This paper is old --- not the first paper on the topic however, that title goes to Manabe and Broccoli (1985) --- and they used a very simple circulation model compared to modern GCMs. We have done a lot of work on this topic in the last few years, and at least a few of these paper ought to be mentioned:
Lofverstrom et al. (2014; 2015; 2016), Lofverstrom and Liakka (2016; 2018), Liakka and Nilsson (2010), Liakka (2012), Liakka et al. (2011; 2016), Liakka and Lofverstrom (2018).

**Zonal LGM jet:**
The way this part of the sentence is written is a bit misleading as both the size and location of the Laurentide ice sheet is important for the zonalisation of the North Atlantic jet, see:

Lofverstrom et al (2014, 2016), Lofverstrom and Lora (2017)

Two competing but not mutually exclusive explanations of the North Atlantic jet zonalisation were provided by:
*Increased frequency of cyclonic wave breaking*: Merz et al. (2015)
*Stationary wave reflection*: Lofverstrom et al. (2016), Lofverstorm and Lora (2017)

**Weaker North Atlantic LGM stormtrack:**
Donohoe and Battisti (2009), Riviere et al. (2018)

**Wave breaking and internal atmospheric variability:**
Ullman et al (2014) did not investigate wave breaking or internal atmospheric variability. A few more appropriate references:
Merz et al (2015), Lofverstrom et al. (2016), Lofverstrom and Lora (2017), Riviere et al. (2018)

Page 2, line 18:
Reconstructions -> Climate model simulations

Page 2, line 21:
Lofverstrom and Liakka (2018) showed that climate biases due to the model resolution can be suppressed (at least to a degree) by the choice of SMB parameterisation.

Page 2, line 22:
Liakka and Lofverstrom (2018) should be mentioned here as well.

Page 3, line 13 and elsewhere:
Neale et al. (2013) is generally considered the "go to" peer-reviewed reference for CAM4

Page 4, line 3:
A nominal 1-degree resolution is generally considered to be an intermediate resolution for climate models (FV1 is the operational resolution of CCSM4). The grid spacing has to be at least 0.25-degrees to be considered high resolution by modern standards.

Page 4, line 7:
Reference showing this?

Page 5, section 2.2:
Are you using the PI land-ocean configuration in all NorESM experiments (i.e. not updating the land-ocean mask based on the ice sheet geometry in PISM)? If so, how is this influencing the results/conclusions? Otto-Bliesner et al. (2016) showed that even relatively small changes in the ocean gateways through Bering Strait and the Canadian Arctic Archipelago can have large implications for the global climate due to changes in the salinity flux in and out of the Arctic basin and into the North Atlantic. Their results were based on Pliocene experiments with CCSM4 (which is using a different ocean component than NorESM), but the climate impact of these ocean gateways is explained by a physical process that realistically should translate to other time periods (and climate models) as well. Regardless, this choice of boundary condition is a huge step away from reality and should therefore be carefully motivated, and potential implications should be discussed.

Page 5, line 17-18:
Cook and Held (1988) used a low-resolution, dry, linear, primitive equations model, which is highly simplistic compared to modern GCMs. See comment above (referring to Page 2, lines 8-12) for more up-to-date references, also regarding the other topics mentioned in this sentence.

Page 5, line 25 and elsewhere:
Sunshine -> insolation (?)

Page 6, line 7:
Do you also change the sub-gridscale topography (fields SGH and SGH30) that are used in the surface drag parameterizations? If not, how is that influencing your results?

Page 6, line 21 and elsewhere:
Probably more accurate to refer to FV1 as an intermediate resolution as it is the default operational resolution of all NCAR models from CCSM4 to CESM2 and many other climate models of the same generation.

Page 7, line 23:
The model used in Cook and Held (1988) didn't have a dynamic ocean.

Page 8, line 9:
The same conclusion was reached in Lofverstrom et al (2014, 2016), Lofverstrom and Lora (2017) and Liakka and Lofverstrom (2018).

Page 10, line 15:
Again, Lofverstrom et al. (2014, 2015, 2016) and Liakka et al. (2011, 2016) showed this as well.

Page 10, line 18:
Assumed by whom? Both Svendsen et al. (2004) and Kleman et al. (2013) suggested that the Eurasian ice sheet had a pronounced east-west extension during the initial phase of the last glacial cycle (first ~20-40 kyrs after the inception), and slowly assumed its LGM configuration with the center of mass in northern Europe.

Page 10, line 20:
We would be careful with the wording here. It is true that these references are discussing "coupled climate--ice-sheet interactions", however the atmosphere model used in these papers is essentially an energy balance model. Many feedbacks are therefore omitted, among them the stationary wave feedback, which is the cornerstone of this study. Lofverstrom and Liakka (2018) discuss potential shortcomnings of climate--ice-sheet interactions in coarse resolution models and models of intermediate complexity, and how these simplifications may help explain some of the systematic biases obtained in the papers referred to here.

Page 10, line 24:
Sentence starting with "Whilst..." is counterintuitive and has not been proven right by these experiments. Ice sheets form where the annual mass balance is positive over long periods of time, i.e. where the net accumulation exceeds the annual mass loss. Mass loss is to first order driven by summer melt (Milankovitch theory), however the climate conditions are also modulated by the atmospheric circulation that can yield warm/cold air advection and turbulent fluxes of sensible and latent heat that can help speed up or slow down ablation/sublimation processes. Similarly, mass gain is proportional to the sum of liquid and solid precipitation. In other words, both mass gain and mass loss are strongly tied to the atmospheric circulation, which in turn is influenced by the presence of ice sheets. It is therefore possible that changes in the circulation induced by growing ice sheets can trigger ice growth elsewhere.

A few papers on how ice sheets are shaped by the mutual interaction with the large scale atmospheric circulation:
Liakka (2012), Liakka et al. (2011), Lofverstrom et al. (2015), Lofverstrom and Liakka (2018)

Page 11, line 1:
Sentence starting with "Once developed..." is not well supported. It may be the case here, but you are also deliberately neglecting changes in several forcing agents (e.g. insolation and

greenhouse gas concentrations) and potentially important feedback loops (e.g. the stationary wave feedback is arguably omitted by letting the ice sheet model run for 100 kyrs before updating the atmospheric state). Also, Lofverstrom et al (2014, 2016), Lofverstrom and Lora (2017) showed that the relative location of the ice sheets in the westerly mean flow is often more important than their size to trigger stationary wave feedbacks.

Page 11, lines 10-14:
Similar questions have been raised and at least partially answered in these papers: Lofverstrom et al (2014, 2016), Lofverstrom and Lora (2017), Liakka et al. (2016), Liakka and Lofverstrom (2018)

Page 11, line 24:
The 100 kyr timescale obtained here is completely artificial and is a direct result of running the ice sheet model for 100 kyrs before updating the atmospheric state. The connection to the "mid-Pleistocene transition" is therefore unjustified.

Fig. 1:
The left panel (MIS4 ice sheet) is never referred to in the text.

**References:**

Donohoe, A., and D. S. Battisti (2009): Causes of reduced North Atlantic storm activity in a CAM3 simulation of the Last Glacial Maximum. J. Climate, 22, 4793–4808, doi:10.1175/2009JCLI2776.1

Liakka, J. (2012): Interactions between topographically and thermally forced stationary waves: Implications for ice-sheet evolution. Tellus, 64A, 11088, doi:10.3402/tellusa.v64i0.11088

Liakka, J, Lofverstrom, M (2018): Arctic warming induced by the Laurentide Ice Sheet topography, Clim. Past, 14, 887–900,https://doi.org/10.5194/cp-14-887-2018

Liakka, J., Lofverstrom, M., and Colleoni, F. (2016): The impact of North American glacial topography on the evolution of Eurasian ice sheet over the last glacial cycle, Clim. Past, 1225–1241, https://doi.org/10.5194/cp-12-1225-2016

Liakka, J., & Nilsson, J. (2010). The impact of topographically forced stationary waves on local ice-sheet climate. *Journal of Glaciology*, *56*(197), 534-544.

Liakka, J., Nilsson, J., and Lofverstrom, M. (2011): Interactions between stationary waves and ice sheets: linear versus nonlinear atmospheric response, Clim. Dynam., 38, 1249–1262

Lofverstrom, M. and Liakka, J. (2016): On the limited ice intrusion in Alaska at the LGM, Geophys. Res. Lett., 43, 11030–11038, https://doi.org/10.1002/2016GL071012

Lofverstrom, M. and Liakka, J. (2018): The influence of atmospheric grid resolution in a climate model-forced ice sheet simulation, The Cryosphere, 12, 1499--1510, https://doi.org/10.5194/tc-12-1499-2018

Lofverstrom, M. and Lora, J. M. (2017): Abrupt regime shifts in the North Atlantic atmospheric circulation over the last deglaciation, Geophys. Res. Lett., 44, 8047–8055, https://doi.org/10.1002/2017GL074274

Lofverstrom, M., Caballero, R., Nilsson, J., and Kleman, J. (2014): Evolution of the large-scale atmospheric circulation in response to changing ice sheets over the last glacial cycle, Clim. Past, 10, 1453–1471, https://doi.org/10.5194/cp-10-1453-2014

Lofverstrom, M., Liakka, J., and Kleman, J. (2015): The North American Cordillera – An impediment to growing the continent-wide Laurentide Ice Sheet, J. Climate, 28, 9433–9450

Lofverstrom, M., Caballero, R., Nilsson, J., and Messori, G. (2016): Stationary wave reflection as a mechanism for zonalising the Atlantic winter jet at the LGM, J. Atmos. Sci., 73, 3329–3342, https://doi.org/10.1175/JAS-D-15-0295.1

Manabe, S. and Broccoli, A. (1985): The influence of continental ice sheets on the climate of an ice age, J. Geophys. Res., 90, 2167–2190

Merz, N., C. C. Raible, and T. Woollings (2015): North Atlantic eddy-driven jet in interglacial and glacial winter climates. J. Climate, 28, 3977–3997, doi:10.1175/JCLI-D-14-00525.1

Neale, R. B., J. Richter, S. Park, P. H. Lauritzen, S. J. Vavrus, P. J. Rasch, and M. Zhang (2013): The mean climate of the Community Atmosphere Model (CAM4) in forced SST and fully coupled experiments. J. Climate, 26, 5150–5168, https://doi. org/10.1175/JCLI-D-12-00236.1

Otto-Bliesner, B.L., Jahn, A., Feng, R., Brady, E.C., Hu, A., Lofverstrom, M. (2017): Amplified North Atlantic warming in the late Pliocene by changes in Arctic gateways. Geophys. Res. Lett. 44, 957–964. http://dx.doi.org/10.1002/2016gl071805

Rivière, G., Berthou, S., Lapeyre, G., and Kageyama, M. (2018): On the reduced North Atlantic storminess during the last glacial period: the role of topography in shaping synoptic eddies, J. Clim., 31, 1637–1652

---

## Author Comment (AC1) · 4 Aug 2018

We would like to use this opportunity to thank Marcus and Johan for reading and commenting our paper. We will take the comments/suggestions into account in our revised version.

We have read most of the paper written by Marcus and Johan. In particular, "The North American Cordillera-An impediment to growing the continent-wide Laurentide Ice Sheet" is very important to explain why coupled models can not simulate a large Laurentide Ice Sheet as reconstructed. We do agree that the ice sheet – stationary wave feedbacks have been studied in recent years. However, our paper is the first to point out how a NE Siberian ice sheet amplifies atmospheric stationary waves, leading

to surface warming in the North Pacific, ablation of the NE Siberian ice sheet, and ultimately a swing to the Laurentide-Eurasian configuration.

Although Marcus and Johan criticize our major conclusion for it being highly artificial and a direct from idealized experiments, we argue that our study is an important step to reconsider the well-established idea about ice sheet developments during glacials. Earlier modelling and geological studies have revealed that a large NE Siberian ice sheet is not impossible. If we do not neglect these studies, we have to investigate the mechanism behind the waxing and waning of the NE Siberian ice sheet. Our study is the fundamental step to distinguish the internal ice sheet – stationary wave feedbacks to the external climate variations (caused by changes in orbital and/or greenhouse gas forcings) in the mechanism.

As we already write in our paper, when we consider both our new modelling results and the existing geological evidence for the history of the NE Siberian ice sheet, we believe the established idea of a gradual expansion into the Laurentide-Eurasian ice sheet configuration has to be revaluated. Our simulations neglect some interactions between the ice sheet and climate variabilities on an orbital timescale, since we run the model to equilibrium with a fixed climate forcing at each step. It still raises some fundamental questions about glacial climate, not least: why was NE Siberia glaciated during some glacials and not others? The internal atmosphere-ocean-vegetation-ice sheet feedbacks highlighted in this study seem to be the key to answer this question. How orbital forcings influence the internal feedbacks to lead to the presence or absence of the NE Siberian ice sheet in past glacials is a question for the future as the possibilities for running transient simulations.

---

## Referee Comment (RC1) · A. Robinson (Referee) · 8 Aug 2018

This study examines the possibility of growth of a large glacial ice sheet in Siberia, and the mechanisms behind it. The authors perform a unique set of simulations of the NH ice sheets asynchronously coupled to a GCM that represents atmospheric stationary waves well. They conclude that indeed certain glacial configurations favor growth of a Siberian ice sheet, which then leads to conditions that favor its demise. This is an interesting study, and certainly serves to advance our understanding of continental-scale ice-sheet dynamics. I recommend publication with only minor revisions, as described below.

First, several feedbacks were mentioned in the introduction and experimental design

[Figure]

(so-called "ice-vegetation-atmosphere-ocean dynamics"), however not much analysis of the vegetation effect was provided other than a few sentences in the Discussion, nor any figures. Does the growth of tundra versus something else mainly cause a temperature impact via albedo? What drives the growth of different vegetation biomes to start with?

Second, is there some justification for the Pb-experiment topography changes, besides that it helps to grow the Barents ice sheet (like this is closer to what the glacial topography would actually look like)? Or is it a correction for model bias? Since the main figures all rely on Pb experiments, this should be made clear.

Third, some description of the simulations using GRISLI, as well as a brief model description should be added more clearly. Now references to these aspects of the study seem very ad hoc and out of place.

Finally, I was also surprised to see the omission of several key papers from the literature. I expect this will be improved following the short comment already posted.

== Minor comments =====

Page 1, line 19: has been => is [?]

Page 1, lines 1-2: Northeast and Northwest should be lowercase. Also, I believe NE and NW are standard abbreviations, no need for definition.

Page 2, line 12: was => has been

Page 2, line 16: studies => evidence

Page 3, line 18: PISM or GRISLI?

Page 3, line 23: ocean model or parameterization?

Page 5, line 21: State somewhere that "pi" is for normal topo and "pb" is for the modified Barents Sea.

Page 9, line 12: behind => behind them

Page 9, line 20: "in the order" . . . of what?

Page 10, line 25: this sentence is not so clear. The feedbacks allow switching from on ice sheet configuration to another. But what determines in the first place whether you will start with the circum-Arctic or Laurentide-Eurasian configuration? Consider revising.

Page 11, line 20: Already said this way, consider rephrasing.

Page 11, line 20: neglect => fail

————————————————

---

## Author Comment (AC2) · 11 Aug 2018

Thanks for reading and commenting our manuscripts. We will take your suggestions into account in the revised version. Here, we provide some quick replies to your questions.

In our experiments, the vegetation modification from taiga forests to tundra leads a strong cooling in NE Siberia. We show this in the supplementary figures. Forced by the change in orbital parameters and the decrease of atmosphere $CO_2$ level, a cooling leads to a vegetation degradation to tudra, which further causes albedo feedback and favours ice sheet growth on NE Siberia.

In the manuscript, we only show simulated results from two-group of experiments, the

[Figure]

Pi group with the modern land-sea distribution, and the Pb group with the Barents Sea closed. All results from these two groups are illustrated in the supplementary materials. Moreover, we have tested many experiments with the land-sea distribution modified. For example, we once closed the Bering Strait, and exposed Arctic continental shelf in our experiments. However, the changes in land-sea distribution do not influence the major conclusion found in the paper.

In the revised the version, we will add the introduction for GRISLI, and the references needed.

---

## Referee Comment (RC2) · Anonymous Referee #2 · 6 Sep 2018

The authors performed simulations of ice sheets under typical glacial conditions using a climate-vegetation-ice sheet model where climate and ice sheets components are coupled asynchronously. The authors found two fundamentally different ice sheet configurations and speculated that spontaneous transitions between these configurations might have happened during glacial cycles. However, I believe that this rather surprising result is an artefact of the flawed methodology employed in this study and therefore it cannot be applied to the real world. This is why bellow I discuss only the methodological aspects of this study.

The authors introduced their model as "state-of-the-art high complexity earth system model" and consider this fact as the "unprecedented advantage" (p. 3) compared to the previous works made with simplified climate models. However, the appropriateness

of the model for the purpose of the study is defined by its most crucial component(s) rather than complexity of the most complex ones. Arguably, for modeling of climate-ice sheet interaction, correct simulation of the surface mass balance (SMB) of ice sheets is of fundamental importance. The authors used a complex but rather coarse-resolution climate model. With its spatial resolution of ca. 400 km, the most important part of ice sheets – the ablation zone - cannot be properly resolved. This is why the accuracy of climate fields simulated over ice sheets depends on the elevation corrections. The authors corrected temperature by using a constant lapse rate and precipitation is assumed to be proportional to the exponent of temperature. Such an approach has been used at least since 90th, mostly for modeling of the Greenland ice sheet response to future climate change. For the centennial timescale and not too strong climate change, this approach can be somehow justified. However, the applicability of this approach to modeling of glacial cycles is questionable at best. In particular, this simple elevation correction does not account for the ice-albedo feedback and therefore cannot be used for modeling ice sheet configurations substantially different from that has been used in climate simulation. Even worse is the situation with precipitation. Precipitation field over ice sheet is strongly topographically controlled and changes in precipitation patterns due to changes in ice sheet extent and elevation cannot be captured by simple temperature correction. Even more in conflict with claimed "high complexity" is the use of the Positive Degree Day method to calculate surface melt. Not only this method is oversimplistic, its inappropriateness even for modeling of the Greenland ice sheet has been shown in numerous studies (e.g. van de Wall, 1996; Bougamont et al., 2007; van de Berg et al., 2011). Equally questionable is using of equilibrium approach for modeling of ice sheets evolution during glacial cycles since the timescales of ice sheets response to orbital forcing are close to periods of orbital forcing.

All these problems significantly affect the reliability of modeling results but, admittedly, they are not unique for the study by Zhang et al. Unfortunately, there is one aspect of this study, namely the way how the asynchronous coupling is implemented, which I believe is fatal for the findings presented in the manuscript. This problem is related to

the implementation of asynchronous coupling technique. Asynchronous coupling has a long history in climate modeling (Manabe and Bryan, 1969). This method is based on the assumption that model results are not seriously affected if the fast component of the system (in this case climate component) is run only a fraction of the full integration time of the slow component (ice sheets). This allows one to reduce significantly computational cost. The method is rather flexible but two conditions have to be met: (i) simulation time of the fast component should be sufficient to allow this component to adjust to the evolution of the slow component; (ii) the periodicity of coupling between fast and slow components should be small enough to ensure that changes of the slow component between coupling events are small enough. If these conditions are met, one can expect that the solution obtained with asynchronous coupling is close enough to that would be obtained with synchronous coupling. For modeling of climate-ice sheet interaction, I would guess that the duration of 100 years for each AOGCM run and coupling between climate and ice sheet components every 1000 years would be a reasonable choice. By the way, such choice would require roughly the same amount of computational time to reach an equilibrium state as the authors actually used. However, for the reason the authors did not explain, they used coupling periodicity of 100,000 years instead. As the result, most of the time (see Fig. 3) the ice sheets evolve under the influence of climate forcing which has been computed for the completely different ice sheets configurations and the simple elevation/temperature correction technique cannot handle this problem. The consequence of this problem is seen in Fig. S2 and S3 (why these two crucial for understanding figures are hidden in SI is not clear to me). After several iterations, the model began to "oscillate" between two completely different ice sheet configurations: "Laurentide-Eurasian" and "circum-Arctic". For both states, the ice sheets and climate are inconsistent with each other as clearly seen in Fig. 3. For example, climate computed for the circum-Arctic configuration leads to a rapid (several thousand years) melting of the huge Eastern Asian ice sheet while climate computed for the Laurentide-Eurasian configuration leads to a rather fast (ca 10 ka) buildup of the Eastern Siberian ice sheet. None of these two ice

sheet configurations can emerge in the real world because this result solely originates from the inappropriate coupling technique. This is why I must conclude that the "instability" of the East Siberian ice sheet found by Zhang and co-authors is just a numerical instability which has nothing to do with the instability of real ice sheets. The closest analogue for such sort of instability is the "checkerboard instability" which arises in the ocean or atmosphere models if the time step is inappropriately chosen.

I realize that the authors put considerable efforts in preparation of the manuscript. However, because this study is based on flawed methodology, the manuscript cannot be simply "revised". If the authors believe that the role of atmospheric stationary waves for the evolution of the Eurasian ice sheet during glacial times has not been yet properly investigated in the previous studies, they should redo all simulations using correct methodology.

———————————————

---

## Author Comment (AC3) · 8 Sep 2018

We are not surprised, but appreciate the tough review on challenging the methodology in our study.

Although the reviewer criticizes the method used to carry out ice sheet modelling, he/she admits the method is widely used.

More critically, the reviewer challenges the asynchronous coupling method with the equilibrated ice sheet simulations. We do not hide this weakness, but point it out in our paper. "In reality, climate forcings evolve alongside the ice sheet so, in an ideal world, fully coupled transient simulations (Ganopolski et al., 2010; Beghin et al., 2014) would most reliably mimic the interaction between climate and ice sheet. However

in a transient simulation, due to the complexity of a fully coupled system, it is quite difficult to distinguish the importance of orbital forcings, greenhouse gas levels and internal ice-vegetation-atmosphere-ocean feedbacks. With constant climate forcing, our asynchronous simulations provide a wealth of new and valuable information, which represents the first step in beginning to fully assess the question of the internal feedbacks and ice sheet configurations. Whilst climate and ice sheet interactions may influence the resulting ice sheet volume and shape, they are unlikely to change whether or not an ice sheet actually forms in NE Siberia or North America. Once developed, ice sheets tend to continue to grow during a full glacial, so the NE Siberian or Laurentide ice sheets should do exactly that, until they become large enough to trigger a swing to another ice sheet configuration. Seen in this way, before a fully coupled ocean-atmosphere-vegetation-ice sheet high complexity model (with a good ability to simulate atmospheric stationary waves) is really available for transient glacial-interglacial simulations, asynchronous coupled simulations remain a good choice to reveal the importance of atmosphere-ocean-vegetation-ice sheet feedbacks in the swings in ice sheet configurations. "

We have rerun asynchronous coupling experiments forced by variable orbital and greenhouse gas forcings, with the ice sheet model running for a few thousand years, during the past glacial-interglacial cycles. We do find the similar result that the NE Siberian ice sheet is unstable.

We know that the idea of gradual development of the Laurentide-Eurasian ice sheet configuration (without the NE Siberian ice sheet) during glacials has been established for more than three decades. However, several climate models can simulate ice sheets over NE Siberia. Our model can produce the numerical instability of the NE Siberian ice sheet. Together with some pieces of geological evidence supporting the NE Siberian ice sheet, it is important for our scientists to be open-minded, to reconsider the possibility of the NE Siberian ice sheet and rethink if the "well-established" idea is really right. It is clear that our study is the fundamental step to reassess this problem.

---

## Author Comment (AC4) · 17 Sep 2018

Dear Editor and Reviewers,

Since we believe that our study is a fundamental advance for understanding ice sheet evolution during past glacial-interglacial cycles, we once tried to submit our paper for reviews on Nature, Science Advance and EPSL. We got a few round of reviews. Some reviewers criticized the equilibrated simulations in our study, and some reviewers did not like our non-mainstream view about NH ice-sheet evolution. However, we would like to use this opportunity to thank all reviewers. All reviewers' constructive suggestions and tough criticisms illuminate us to carry out new studies to make our conclusion more robust. Here, we would like to summarize the major criticisms, and give them replies.

[Figure]

Criticism 1. The appearance of NE Siberian ice sheet is due to the cold biases of CAM atmosphere model. The CAM outputs were used directly in the PISM simulations. It is likely the cold biases that cause an ice-sheet simulated over NE Siberia.

This is not case. Although CAM could have some cold biases in the simulated surface air temperature, our experiment show that an ice sheet can not grow on NE Siberia without changes in vegetation cover there. It is the vegetation-albedo feedback causes a strong cooling over NE Siberia, allow an ice sheet grow there. As Marcus Lofverstrom and Alexander Robinson mentioned in their comments, "the vegetation feedback is often omitted in climate modelling as reliable reconstruction for glacial condition remain elusive – e.g. preindustrial vegetation is specified for the LGM in the PMIP1-4 boundary conditions – make this study somewhat unique."

Criticism 2. As pointed out by the Anonymous Referee, the swings of two ice sheet configurations found in this study are based on the idealized experiments, with equilibrated ice sheet simulations. Although a NE Siberian ice sheet is not totally impossible, in reality (or a fully coupled model system), it is likely that an ice sheet can not grow large on NE Siberia, due to it's warming feedback. In a precession cycle, the short duration could also limit the growth of ice sheet on NE Siberia. Since the NE Siberian can not grow large, it can not trigger the swing to the Laurentide-Eurasian configuration.

We acknowledge that, based on the experimental design in the current study, it is difficult for us to answer this criticism. However, we do carry out new simulations with much short time steps for ice sheet model, we do find the similar result that the NE Siberian ice sheet grows large, and is unstable. In the current study, although we run the PISM ice sheet model to an equilibrium, the time series in Fig.4 show that, with the ice sheet model running for 4000~6000 years, the NE Siberian ice sheets reach ~1500 m high, which is high enough to influence atmospheric stationary waves.

Criticism 3. The simulated influence of ice sheets on atmospheric stationary waves is model-independent, and remain uncertainties.
We acknowledge this criticism. We once tried to impose a NE Siberian ice sheet in the IPSL atmosphere model. The IPSL does not give similar responses as we simulated with CAM in the current study. A modelling study is always model-independent. However, it does not mean the simulated ice sheet-stationary waves feedbacks are wrong in our study. Based on the modern climate study, it seems that CAM has a better ability in simulating atmospheric stationary waves.

Actually, the "well-established" idea about the gradual enlargement of the Laurentide-Eurasian configuration during glacials is not irrefutable. Some geological evidence does support that NE Siberia was once glaciated. Despite new evidence continuously arise, hinting at the occurrence of an ice sheet in this region, the idea of NE Siberia ice sheet is discounted in favour of a focus on the dominance of only one ice sheet configuration. Some geological and modelling studies suggest that NE Siberian ice sheet was once large during MIS6, which is definitely against the well-established idea. If NE Siberia can be glaciated during the penultimate glacial, why it must be unglaciated during the last glacial?

The evidence provided in our current study is not strong enough to challenge the "well-established" idea. However, it opens a new window to rethink if the "well-eastablished" idea is really right. Our asynchronous coupling method, with constant climate forcing, although criticize by reviewers, highlights vital ice-vegetation-ocean-atmosphere feedbacks, – something that is not be possible for previous transient simulations. The mechanism revealed in this study is very likely the key for reconsidering the complex ice-sheet development during past glacial-interglacial cycles.

Regards

Zhongshi Zhang on behalf of all co-authors
* * *